

# Field-induced local Fermi-liquid states in the underscreened Kondo model for Eu compounds

**Shingo Kuniyoshi[1] and Ryousuke Shiina[2]**

**1** Graduate School of Engineering and Science, University of the Ryukyus, Okinawa 903-0213, Japan
**2** Faculty of Science, University of the Ryukyus, Okinawa 903-0213, Japan

⋆ shiina@sci.u-ryukyu.ac.jp

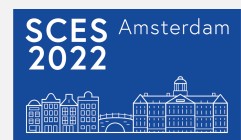 *International Conference on Strongly Correlated Electron Systems (SCES 2022)*
## Abstract

We have studied an underscreened Kondo model for the impurity spin $S = 7/2$ capturing the $4f$ characteristics of a nearly divalent Eu ion. By using the numerical renormalization group method, it is shown that an unusual heavy-fermion state is induced from the underscreened spin state when a weak crystal field is introduced. It is also found that the application of the magnetic field enhances the effective mass of the heavy fermions, reflecting the field-induced level crossing of the spin states. We will discuss the mechanism of these mass enhancements in terms of a realization of the two-stage Kondo effect.

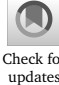
# 1   Introduction

Recent experimental activities have definitely clarified a rich variety of low-temperature properties of the Eu compounds; one can now encounter exotic antiferromagnetic orders, quantum critical phenomena, heavy fermions (HF), and valence transition or crossover phenomena in several Eu compounds, when controlling the pressure and/or the doping rate [1–9]. As a basis to clarify the microscopic origin of such interesting physics of the Eu system, we have introduced the Eu-based impurity Anderson model in a previous paper [10] and have shown that the total singlet ground state is realized in the entire region between the divalent and trivalent Eu ion. It should be stressed, however, that the mechanism and the properties of the HF states in the Eu systems remain to be open. In this paper, we shall investigate the HF formation in the divalent limit and point out that it shows marked difference from the conventional one which has often been observed in the Ce compounds.

The dominant valence state of $Eu^{2+}$ consists of the half-filled $4f$ shell, where a pure spin state $S = 7/2$ is formed by the Hund's rule coupling with quenching the orbital angular momentum. A small hybridization between the $4f$ and conduction electrons gives rise to a $c$-$f$ exchange interaction and then contributes partial screening to the large $4f$ spin. On the other hand, as shown in the previous paper [10], the spin-orbit coupling induces an anisotropic Kondo coupling and a crystal field, which can complement the singlet formation. Thus, the $4f$ state of the divalent Eu ion is characterized by the underscreened Kondo (USK) coupling with small magnetic anisotropy.

In this paper, we study the effects of crystal and magnetic fields on the HF formation in the USK model by the numerical renormalization group (NRG) method [11]. We first confirm that an unconventional HF state under a finite crystal field is induced from the so-called singular Fermi liquid state in the isotropic limit [12, 13], as we have shown in a separate paper [14]. Next, we shall investigate the magnetic field effect on the USK model with a fixed crystal field. It is found that the weak magnetic field hampers the singlet formation owing to the crystal field and enhances remarkably the effective mass of the HF state. The mechanism of the field-induced HF states is discussed in terms of the two-stage Kondo effect.

# 2   Underscreened Kondo model

We shall study the following USK model for the divalent Eu ion,

$$H = \sum_{k\sigma} \varepsilon_{k\sigma} c_{k\sigma}^{\dagger} c_{k\sigma} + J\boldsymbol{\sigma} \cdot \boldsymbol{S} + \Delta S_z^2 + h S_z. \tag{1}$$

Here, the first term is the band energy of conduction electrons with the creation operator $c_{k\sigma}^{\dagger}$ for a momentum $k$ and a spin $\sigma$. In the second term, $\boldsymbol{S}$ and $\boldsymbol{\sigma} \equiv \frac{1}{2} c_{\sigma}^{\dagger} \boldsymbol{\rho}_{\sigma\sigma'} c_{\sigma'}$ stand for the impurity spin operator of $S = 7/2$ and the spin-density operator of conduction electrons, respectively, where $\boldsymbol{\rho}$ means the Pauli matrix. The coupling constant $J(> 0)$ is assumed to be antiferromagnetic, because it is derived from the perturbative valence fluctuation from the $4f^7$ to $4f^6$ configurations [10]. We assume the crystal field with $\Delta(> 0)$ in the third term and the magnetic field in the fourth term, both of which are applied only to the impurity spin.

It is well known that the interaction grows to the strong coupling, in the sense of the renormalization group, for the temperatures $T < T_K$ at least in the absence of the fields. Therefore, it is instructive to consider the strong-coupling limit of the model; it is reduced to a mere sum of free electrons and the decoupled $\tilde{S} = 3$ spin,

$$H_{\mathrm{dcf}} = \sum_{k\sigma}{}' \varepsilon_{k\sigma} c_{k\sigma}^{\dagger} c_{k\sigma} + \tilde{\Delta} \tilde{S}_z^2 + \tilde{h} \tilde{S}_z. \tag{2}$$

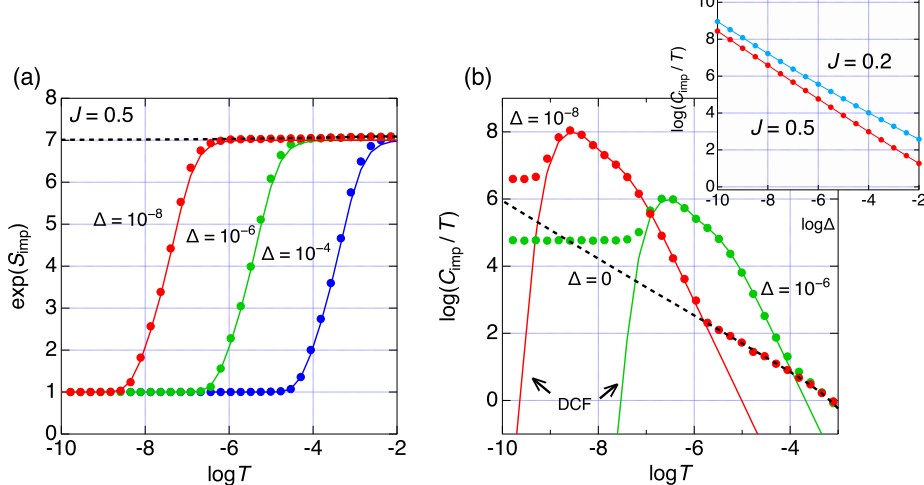

Figure 1: The temperature dependences of the exponential of the entropy $S_{\text{imp}}$ and the logarithm of the specific heat coefficient $C_{\text{imp}}/T$ with fixing $J = 0.5$ are shown in (a) and (b), respectively. The dashed line and symbols represent the results of the USK model (1) for $\Delta = 0$ and $\Delta = 10^{-4}, 10^{-6}, 10^{-8}$, respectively. For comparison, the results of the DCF model (2) are plotted by lines for the same parameters. The inset in (b) shows the $\Delta$ dependence of $C_{\text{imp}}/T$ at the lowest temperature for $J = 0.2$ and 0.5.

Here the prime on the $k$ sum means the presence of a defect at the impurity site. Note also that the crystal field and the Zeeman energy are renormalized to be $\tilde{\Delta} = 5\Delta/4$ and $\tilde{h} = 9h/8$, respectively, for the $\tilde{S} = 3$ spin state, where the factor is derived from the Clebsch-Gordan coefficient with $S = 7/2$ and $\sigma = 1/2$. We shall call it the decoupled crystal field (DCF) model in the following.

In the case of a finite crystal-field energy, one might naively expect that the spin degeneracy is removed with $\Delta(> 0)$ and the local singlet with $\tilde{S}_z = 0$ becomes the ground state. It should be noted, however, that the approach to the strong-coupling fixed point is extremely slow in the USK model, owing to the emergence of a marginally irrelevant interaction between the composite spin and conduction electrons. As a result, the model in the absence of the fields is known to display an apparent non-Fermi-liquid thermodynamics at low temperatures, which has sometimes been called the singular Fermi liquid (SFL) [12, 13].

## 3 Numerical results

In this study, we have applied the NRG method [11] to the model (1). The discretization parameter $\Lambda = 3$ is used with 700-2500 states kept at each renormalization step. We have confirmed that the following numerical results converge sufficiently within the number of the states. We have also checked that $\Lambda = 2$ gives the same results with slower convergences. The details of the NRG setup are essentially the same as in Ref. 10. Note that the half-bandwidth is set to $D = 1$ to be used as the energy unit in the results.

First, the numerical results for the entropy with fixing $J = 0.5$ are shown as a function of the temperature in Fig. 1(a). In the case of $\Delta = 0$, the entropy takes $\ln 7$ at low temperatures that corresponds to the composite spin state $\tilde{S} = 3$. Since the Kondo temperature is larger than $T = 10^{-2}$ for $J = 0.5$, the crossover from $S = 7/2$ does not appear in the temperature region

of Fig. 1(a). When $\Delta$ is introduced, the entropies are not changed at high temperatures, indicating the formation of $\tilde{S} = 3$. On the other hand, it is found that they commonly exhibit distinct reductions roughly below $T \sim \Delta$. For comparison, we also plot the results of the DCF model and find that they seemingly reproduce the results of the USK model with finite $\Delta$. This agreements indicate at least that the predominant entropy changes at low temperatures are brought about by crystal field splitting, as suggested by several authors [15–17]. It is to be stressed, however, that these agreements are not so clear in the asymptotic slopes of the two models in the low-temperature limit. Thus, a careful study is necessary to identify the exact mechanism of the singlet formation.

For this purpose, we shall discuss the temperature dependence of the impurity specific heat $C_{\text{imp}} \equiv C - C_0$, where $C$ and $C_0$ represent the specific heat of the USK model (1) and that of the free conduction band, respectively. In Fig. 1(b), we plot the coefficient of the specific heat $C_{\text{imp}}/T$ for several values of $\Delta$ with fixing $J = 0.5$. In the case of $\Delta = 0$, it is shown that the NRG result exhibits a divergence with decreasing the temperatures, whose asymptotic form coincides with the known analytic result of the SFL $C_{\text{imp}}/T \propto T^{-1} \ln^{-4}(T_K/T)$ [12, 18].

When $\Delta$ is introduced, the coefficients definitely depart from the SFL behavior of $\Delta = 0$ at intermediate temperatures and become consistent with those derived for the DCF model in the low-temperature region. It is remarkable, however, that $C_{\text{imp}}/T$ turns into a constant with lowering the temperature further, contrary to the speculation from the gross feature of the entropy. Clearly, it is nothing like an exponential reduction of the DCF model in the same region. Note that we have found a local Fermi-liquid behavior also in the spin susceptibility $\chi_{\text{imp}} = \partial \langle S_z \rangle / \partial h$, though not showing in the figure; it exhibits a fixed value independent of the temperatures in the low-temperature limit, as well. Thus, it is found that the USK model with $\Delta \neq 0$ results in the three representative temperature regimes; the high-$T$ SFL state and the subsequent DCF state are characterized by $T_K$ and $\Delta$, respectively, whereas the low-$T$ Fermi liquid state can be affected by both.

Next, we shall discuss the numerical results for the $\gamma$ value, namely, $C_{\text{imp}}/T$ at the lowest temperature, as a function of $\Delta$ in the inset of Fig. 1 (b). We have confirmed that $C_{\text{imp}}/T$ becomes constant for each parameter set at low temperatures. It is shown that the $\gamma$ value is significantly enhanced when $\Delta$ is small, seemingly in proportion to $\Delta^{-1}$. In addition, it is found to be enhanced further with decreasing $J$ when $\Delta$ is fixed. In the previous paper, we have shown that the NRG results of $\gamma$ are quantitatively consistent with the analytical expressions derived by the Large-$N$ bosonization theory, although the present case corresponds to the case of the smallest $N = 2$ in the theory [12, 14].

Next, we investigate the magnetic field effect in the USK model. The numerical results of the entropy with fixing $J = 0.5$ and $\Delta = 10^{-4}$ are shown as a function of the temperature in Fig. 2 (a). We found that the entropies are enhanced at low temperatures when the field $h$ is introduced. It is particularly remarkable that a plateau-like structure of the entropies appears in the low-temperature region, when $h \sim \Delta$. In fact, the emergence of a low-lying doublet is expected by field-induced level crossing between $\tilde{S}_z = 0$ and $\tilde{S}_z = 1$ in the DCF model. If it is the case, the singlet formation around the level-crossing field at low temperatures is brought about by the second Kondo effect, apart from the first one with a much higher Kondo temperature. Thus, it can be a variant of the so-called two-stage Kondo effect [16, 17]. We found that the crossover temperature from the doublet to the singlet in the entropy seems to take a minimum at $h^* = 1.086 \times 10^{-4}$, where the crossover is comparatively slow. On the other hand, when $h$ is larger than $\Delta$, the entropy decreases abruptly from $\ln 7$ around $T = h$, without showing the $\ln 2$ plateau.

We shall discuss the corresponding results for $C_{\text{imp}}/T$ in Fig. 2 (b). When the field is applied, a new peak in $C_{\text{imp}}/T$ appears in the low-temperature side, probably due to the Zeeman splitting of the local excited states. As the temperature is lowered below the low-temperature

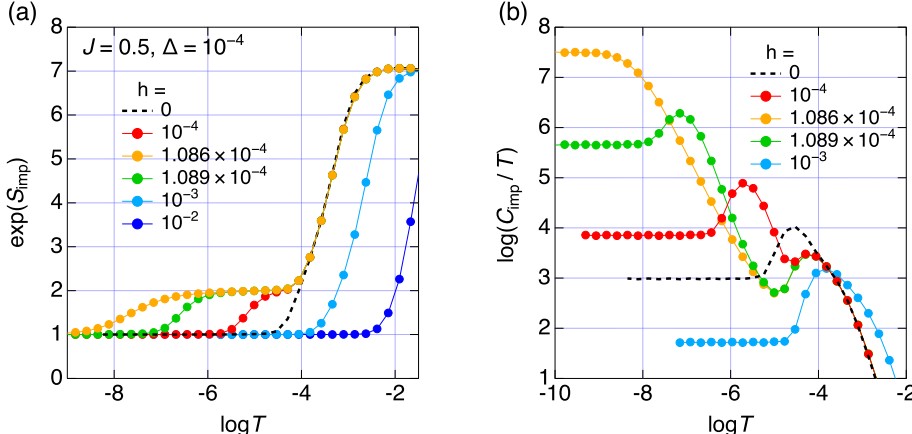

Figure 2: The temperature dependences of the exponential of the entropy $S_{\text{imp}}$ and the logarithm of the specific heat coefficient $C_{\text{imp}}/T$ with fixing $J = 0.5$ and $\Delta = 10^{-4}$ are shown in (a) and (b), respectively. The dashed line and symbols are the results of the USK model for several values of $h$.

peak, one can again find the crossover to the Fermi-liquid state with a constant $\gamma$. The $\gamma$ values are found to be enhanced remarkably as the field is increased, being opposite to the well-known feature of the normal Fermi liquid in the $s = 1/2$ Kondo model. On the other hand, it is also found that the low-temperature peak structure disappears and the $\gamma$ value becomes the largest at $h = h^*$. These behaviors for $h \sim \Delta$ are consistent with the two-stage Kondo scenario mentioned above. When $h$ increases further, the $\gamma$ value is rapidly suppressed and becomes even smaller than the value at zero field.

We have also calculated the $\gamma$ value as a function of the magnetic field. We have found that it shows a sharp peak structure at $h = h^*$ without a divergence. When one focuses on the fine structures around $h \sim \Delta$, one can find additional two peaks at field strengths larger than $h^*$, which are expected to come from level crossing of the ground state with $\tilde{S}_z = 2$ and 3, respectively. The numerical results and the detailed interpretations of the fine structures will be discussed elsewhere.

## 4 Conclusion

In this paper, we have reported the NRG results of the USK model with the crystal and magnetic fields. It was shown that the SFL state in the USK model is transformed to the local Fermi-liquid state by a crystal field, which opens up a possibility of realizing the HF state even in the divalent Eu ion. The characteristic energy scale of the HF state is given by the combination of the crystal field and the Kondo coupling, predicting that the mass enhancement favorably takes place for a nearly isotropic system with a small crystal field. It was also found in this paper that the HF behaviors are further enhanced by applying the magnetic field in a finite crystal field, contrary to the field effect in the conventional HF state. We believe that these results can provide a basis to understand the complex experimental results of the Eu compounds. Note, finally, that the present results may also be useful for the HF physics in other rare-earth systems with a sufficiently small crystal field.

## Acknowledgements

This work was partially supported by JSPS KAKENHI Grant Number 19K03735. S. K. thanks to a support from the fund for SCES2019.

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
