# Peer review of "Field-induced local Fermi-liquid states in the underscreened Kondo model for Eu compounds"

_SciPost Physics Proceedings, doi:SciPost Phys. Proc. 11, 015 (2023)_

## Round 1 · Referee Report · Anonymous (Referee 1) · 2023-2-22

Strengths

  • new interesting numerical results on the underscreened Kondo model in a weak crystal and magnetic field, building upon two previous works by the authors

Weaknesses

  • discussion of the numerical errors is missing

Report

In this paper the authors use the numerical renormalization group (NRG) to study an underscreened Kondo model for the impurity spin S=7/2 in a weak crystal and magnetic field, relevant for Eu-based compounds. They show that a finite crystal field transforms the non-Fermi liquid state in the isotropic limit into an unconventional heavy-fermion state. They find that the effective mass of the heavy fermions is strongest for a nearly isotropic system with a small crystal field, and that a magnetic field further enhances it, whose mechanism they explain in terms of a two-stage Kondo effect.

This paper builds upon two previously published papers by the authors and extends them by considering also the case of a magnetic field. In my opinion this work provides an interesting and valuable contribution to the ongoing efforts in understanding the heavy-fermion physics of EU-based compounds (and potentially other rare-earth systems), and I can thus recommend this article to be published in SciPost Physics Proceedings.

There is one point which the authors should improve. The information on the numerical approach in the paper is very short. I agree that it is not necessary to include all the details, if the setup is similar to the one used in Ref.10, but it is nevertheless important to comment on the numerical errors in the data. For example, is all data sufficiently converged as a function of the number of states kept, and is the dependence on the discretization parameter negligible? I assume this is the case, but it would be important to explicitly state this in the paper.

Requested changes

  • add comment(s) regarding the involved numerical errors

  • validity: good
  • significance: good
  • originality: good
  • clarity: good
  • formatting: good
  • grammar: good

Author:  Shingo Kuniyoshi  on 2023-03-01  [id 3413]

(in reply to Report 1 on 2023-02-22)

Thank you for quick response and the suggestion to improve the paper.
We agree with your comment that discussion of the numerical errors should be included in the paper.
Regarding the errors, we confirmed the following two:
・The numerical results for the thermodynamic quantities sufficiently converge by keeping 2500 states at each renormalization step for the discretization parameter Λ = 2-3.
・The dependence on Λ for the thermodynamic quantities is negligibly small, for example, the results of Cimp/T at the lowest temperature for Λ=3 and 2 only results in a small difference of a few percentage points.

We have revised the first paragraph of the section 3 on the basis of the above aspects.

Once again, we are truly grateful for your comment.

Shingo Kuniyoshi and Ryousuke Shiina.

---

## Editorial Decision

published